# Cohort profile: Epigenetics in Pregnancy (EPIPREG) – population-based sample of European and South Asian pregnant women with epigenome-wide DNA methylation (850k) in peripheral blood leukocytes

Nicolas Fragoso-Bargas[1,2], Julia O. Opsahl[2], Nadezhda Kiryushchenko[1,3], Yvonne Böttcher[2,4,5], Sindre Lee-Ødegård[2], Elisabeth Qvigstad[1,2], Kåre Rønn Richardsen[6], Christin W. Waage[7,8], Line Sletner[2,9], Anne Karen Jenum[8], Rashmi B. Prasad[10], Leif C. Groop[10], Gunn-Helen Moen[2,11,12,13], Kåre I. Birkeland[1,2], Christine Sommer[1] *

1 Department of Endocrinology, Morbid Obesity and Preventive Medicine, Oslo University Hospital, Oslo, Norway, 2 Institute of Clinical Medicine, University of Oslo, Oslo, Norway, 3 Department of Bioscience, University of Oslo, Oslo, Norway, 4 Department of Clinical Molecular Biology, Akershus University Hospital, Lørenskog, Norway, 5 Helmholtz-Institute for Metabolic, Adiposity and Vascular Research, Leipzig, Germany, 6 Faculty of Health Sciences, Department of Physiotherapy, Oslo Metropolitan University, Oslo, Norway, 7 Faculty of Health Sciences, Department of Physiotherapy, Oslo Metropolitan University, Oslo, Norway, 8 Department of General Practice, General Practice Research Unit (AFE), Institute of Health and Society, University of Oslo, Oslo, Norway, 9 Department of Pediatric and Adolescents Medicine, Akershus University Hospital, Lørenskog, Norway, 10 Lund University Diabetes Centre, Malmö, Sweden, 11 The University of Queensland Diamantina Institute, The University of Queensland, Woolloongabba, QLD, Australia, 12 Department of Public Health and Nursing, K.G. Jebsen Center for Genetic Epidemiology, NTNU, Norwegian University of Science and Technology, Trondheim, Norway, 13 Population Health Science, Bristol Medical School, University of Bristol, Bristol, United Kingdom

* christine.sommer@medisin.uio.no

## Abstract

Pregnancy is a valuable model to study the association between DNA methylation and several cardiometabolic traits, due to its direct potential to influence mother's and child's health. Epigenetics in Pregnancy (EPIPREG) is a population-based sample with the aim to study associations between DNA-methylation in pregnancy and cardiometabolic traits in South Asian and European pregnant women and their offspring. This cohort profile paper aims to present our sample with genetic and epigenetic data and invite researchers with similar cohorts to collaborative projects, such as replication of ours or their results and meta-analysis. In EPIPREG we have quantified epigenome-wide DNA methylation in maternal peripheral blood leukocytes in gestational week 28±1 in Europeans (n = 312) and South Asians (n = 168) that participated in the population-based cohort STORK Groruddalen, in Norway. DNA methylation was measured with Infinium MethylationEPIC BeadChip (850k sites), with technical validation of four CpG sites using bisulphite pyrosequencing in a subset (n = 30). The sample is well characterized with few missing data on e.g. genotype, universal screening for gestational diabetes, objectively measured physical activity, bioelectrical impedance, anthropometrics, biochemical measurements, and a

**Data Availability Statement:** Due to strict regulations for genetic data and privacy protection of patients in Norway, all requests for data access are processed by the STORK Groruddalen project's steering committee. Data access requests can be filed to the PI of STORK Groruddalen (a.m.l. brand@medisin.uio.no) or the PI of EPIPREG (christine.sommer@medisin.uio.no).

**Funding:** EPIPREG is supported by the South Eastern Norway Regional Health Authority (grant number: 2019092), and the Norwegian Diabetes Association (grant number: N/A). G.H.M. is supported by the Norwegian Research Council (Post doctoral mobility research grant 287198), and have received funding support by Nils Normans minnegave (grant number: N/A).

**Competing interests:** The authors have declared that no competing interests exist.

biobank with maternal serum and plasma, urine, placenta tissue. In the offspring, we have repeated ultrasounds during pregnancy, cord blood, and anthropometrics up to 4 years of age. We have quantified DNA methylation in peripheral blood leukocytes in nearly all eligible women from the STORK Groruddalen study, to minimize the risk of selection bias. Genetic principal components distinctly separated Europeans and South Asian women, which fully corresponded with the self-reported ethnicity. Technical validation of 4 CpG sites from the methylation bead chip showed good agreement with bisulfite pyrosequencing. We plan to study associations between DNA methylation and cardiometabolic traits and outcomes.

## Introduction

Studies of epigenetic marks have in recent years gained increased interest in the context of human diseases [1]. Such studies may enhance our biological understanding of the aetiology of several diseases, increase our understanding of detrimental or protective mechanisms, or for prognosis and risk prediction [2]. One of the most studied epigenetic mechanisms is DNA methylation, which plays an important role in normal development, chromatin organization and gene expression [3]. Several studies have indicated that DNA methylation is associated with cardiovascular risk factors such as body mass index (BMI) [4–7], gestational diabetes (GDM) [8], type 2 diabetes (T2D) [9–13], lipid levels [14,15], hypertension [16,17], smoking [18–22] and alcohol intake [23–25], suggesting that cardiometabolic diseases have an epigenetic component.

Although scarcely studied, pregnant women provide a unique opportunity to study the association between blood DNA methylation and several phenotypes related to glucose homeostasis and cardiovascular traits. This is because pregnancy has been proposed as a stress test for metabolism in several organs [26], including the pancreatic beta-cells, since insulin resistance increase naturally in all pregnancies [27]. In the third trimester of pregnancy, this insulin resistance in many women reaches a level similar to that observed in type 2 diabetes, requiring the beta-cell to increase its insulin secretion considerably to compensate [28]. Similarly, pregnancy-induced hypertension is associated with increased risk of future cardiovascular disease [29].

In the Epigenetics in Pregnancy (EPIPREG) sample, we have quantified epigenome-wide DNA-methylation in peripheral blood leukocytes in women of European and South Asian origin attending the well-characterized, multi-ethnic and population-based STORK Groruddalen (STORK G) study [30]. The population based design and inclusion of women with Western and South Asian ethnicity allow us to study of a wide range of phenotypes to detect either ethnic-specific and/or common DNA methylation patterns in relation to phenotypes of interest. Furthermore, South Asians are of special interest due to their higher weight retention after pregnancy, increased prevalence of gestational diabetes, and increased risk for later type 2 diabetes compared to Europeans [31,32]. The aim of EPIPREG is to discover novel associations between DNA-methylation in pregnancy and cardiometabolic related traits in South Asian and European pregnant women and their offspring, which may have potential for prevention and treatment. This cohort profile paper aims to present our sample with genetic and epigenetic data and invite researchers with similar cohorts to collaborative projects, such as replication of ours or their results and meta-analysis.

## Cohort description

### Study population

EPIPREG (n = 480) is a sub-study of the larger STORK Groruddalen (STORK G) study, which is a population-based cohort of 823 healthy women with different ethnic origin (European, South Asian, African, Middle Eastern and South American) attending three public Child Health Clinics for antenatal care in the multi-ethnic area of Groruddalen, Oslo, Norway, 2008–2010 [30]. Briefly, women were eligible if they: 1) Lived in the study districts; 2) planned to give birth at one of two study hospitals; 3) were <20 weeks pregnant; 4) could communicate in Norwegian or any of the eight translated languages; and 5) were able to give an informed consent. Women with pre-gestational diabetes, or in need of intensive hospital follow-up during pregnancy, were excluded. The overall participation rate in STORK G was 74%, 81.5% for Europeans and 73% for South Asians [30].

### Ethical approval

The STORK G study including genetic and epigenetic data is approved by the Norwegian Regional Committee for Medical Health Research Ethics South East (ref.number 2015/1035). We obtained written informed consent from all participants before any study-related procedure.

### Data collection

**Questionnaire data and anthropometrics.** In STORK G, interviewer-administered questionnaires were completed at gestational week 15±3 (visit 1) and 28±2 during pregnancy (visit 2), and 12±2 weeks postpartum (visit 3) [30,33]. Questionnaire data included information on mother's general health, physical activity and a dietary habits, in addition to some information about the father. Details about the questions used and data gathered have been described previously [30], and are available upon request to the corresponding.

Ethnic origin was defined by either the individual's country of birth or their mother's country birth, if the latter was born outside Europe [34]. We have detailed data on parity, pre-pregnant BMI, smoking status, alcohol intake, education, marital status and diet [30,33,35]. At all the three visits, we measured maternal height, body weight, fat mass with bioelectrical impedance (Tania-Weight BC-418 MA), skinfold thickness at three sites (Holtain T/W Skinfold Caliper, Holtain Ltd., Crymych) [31] and systolic and diastolic blood pressure (Omron HEM-7000-E M6 Comfort) [36].

**Universal screening for gestational diabetes.** All women underwent a 75 g oral glucose tolerance test at gestational week 28 ±2. Fasting and 2-hour glucose were analysed with a point-of-care instrument (HemoCue, Angelholm, Sweden). Women were diagnosed with gestational diabetes based on the WHO 1999 criteria (fasting glucose $\geq$ 7.0 mmol/l or 2-hour glucose $\geq$ 7.8 mmol/l) [37]. Furthermore, in retrospect and exclusively for research purposes, we also classified the samples using a slightly modified version of the WHO 2013 criteria (fasting glucose $\geq$ 5.1–6.9 mmol/l or 2-hour glucose $\geq$ 8.5–11 mmol/l, no data for 1-hour glucose) [38].

**Laboratory data.** Venous blood was drawn at the three visits into tubes with ethylenediaminetetraacetic acid (EDTA). Subsequently, the samples were aliquoted and biobanked or subject to routine laboratory analyses that were performed continuously during the study period. Fasting glucose, total Cholesterol, LDL-Cholesterol, HDL-Cholesterol and triglycerides levels were measured with a colorimetric method (Vitros 5.1 fs, Ortho clinical

diagnostics) [35], HbA1c levels were assessed in full blood with HPLC (Tosoh G8) [34], fasting C-peptide and insulin were measured at the Hormone Laboratory, Oslo University Hospital, with non-competing immunoflurometric assays (DELFIA, PerkinElmer Life Sciences, Wallac Oy, Turku, Finland) [39]. Serum 25(OH)D was analysed by competitive RIA (Dia-Sorin) at visit 1 and visit 2 [40], and S-leptin was analyzed by HADCYMAG-61K based on Luminex® xMAP® technology [32], at the Hormone Laboratory, Oslo University Hospital. Serum vitamin B12 and folate were measured with Electrochemiluminescence (ECLIA) assays, Roche, at Medical Biochemistry, Oslo University Hospital. HOMA-IR and HOMA-B [39,41] were estimated by Oxford University HOMA Calculator 2.2 using fasting glucose and C-peptide.

**Objectively measured physical activity.** Physical activity (PA) was objectively measured from visit 1 to 3 using SenseWear™ Pro3 Armband (SWA) (BodyMedia Inc, Pittsbur, PA, USA) [42]. Data from women with at least one valid day (defined as $\geq$ 19.2h) were considered valid [43]. Physical activity was characterized as Sedentary behaviour ($<$ 1.5 metabolic equivalents (METs)) light intensity (1.5 to $<$3 METS) or moderate or intense ($\geq$ 3 METs) [42,44].

**Offspring data.** The STORK G study also collected abdominal circumference, head circumference, bi-parietal diameter, femur length and estimated fetal weight by ultrasound measured on three different time points during pregnancy, and gestational age at birth derived from the first day of the woman's last menstrual period [45]. We have detailed anthropometric measurements at birth such as birthweight, head circumference, abdominal circumference, crown-heel length and neonatal skinfolds measured with a Holtain T/W Skinfold Caliper (Holtain Ltd., Crymych) [46]. Measurements of weight and length/height were collected during routine follow-up at the Mother—and Child Health Clinics when the children were 6 weeks old and thereafter at the 3, 6, 12, 15, 24 and 48 months visits. Further register-based follow-up is planned.

Venous serum cord blood samples were collected at birth and stored at -80˚C. Several sections from the placenta and umbilical cord have been sampled, and stored as Formalin-Fixed Paraffin-Embedded (FFPE) blocks. Currently, an ongoing pilot study of 80 FFPE placentas (40 South Asian, 40 European) demonstrate that enough DNA can be extracted from the FFPE to successfully run pyrosequencing (Sletner, unpublished). Furthermore, we have frozen placenta biopsies of about 1/3 of the women's offspring.

**DNA extraction.** In STORK G, at gestational week 28±2, DNA from peripheral blood leukocytes was extracted continuously throughout the data collection, at the Hormone Laboratory, Oslo University Hospital, using a salting out procedure [47], and stored at -80˚C.

**Genetic data.** The samples were genotyped using the Illumina CoreExome chip, by the Department of Clinical Sciences, Clinical Research Centre, Lund University, Malmö, Sweden [48]. Of the 664 genotyped samples those with low call rate (i.e. $<$ 95%, n = 0), extreme heterozygosity ($>$ |mean± (3xSD)|, n = 1), mismatched gender (n = 24) or cryptic relatedness (i.e. one individual (chosen at random) from each related pair, defined as genome-wide Identity by descent (IBD) $>$ 0.185 (n = 6) were excluded from analyses. Genetic ethnic origin was defined by ancestry informative principal component analysis based on the variance-standardized relationship matrix generated in PLINK 1.9 software package [49] (https://www.cog-genomics.org/plink/1.9/)).

Variants with call rate <95% (10081 SNPs), out of Hardy-Weinberg equilibrium (exact $p<10^{-6}$, 1971 SNPs) or with low minor allele frequency (MAF) <1% (245221 SNPs) were removed before imputation. Quality control was performed using the PLINK 1.9 software package [49]. After quality control, 293914 variants were left for imputation.

Imputation in European and South Asian samples was performed as follows: The GWAS scaffold was mapped to NCBI build 37 of the human genome. Imputation to the 1000G reference panel (Phase3- http://www.well.ox.ac.uk/~wrayner/tools/) was performed separately in Europeans and South Asians using their respective panels from the 1000G. The populations that are included in the European panel are Utah residents with Northern and Western European ancestry, Iberian populations in Spain, Finish in Finland, British in England and Scotland, and Tuscany in Italy. In the South Asian panel the included populations are Bengali in Bangladesh, Gujarati Indian in Houston, Texas, Indian Telugu in the UK, Punjabi in Lahore, Pakistan and Sri Lankan Tamil in the UK. The software used for imputation was IMPUTE (version 2.3.2) [50].

**Epigenome-wide DNA methylation.** Europeans and South Asians were the largest ethnic groups in STORK G, and South Asians of special interest due to their higher weight retention after pregnancy, increased prevalence of gestational diabetes, and increased risk for later type 2 diabetes compared to Europeans [32]. In EPIPREG, we quantified DNA methylation in maternal peripheral blood leukocytes in gestational week 28±1.2 in all Europeans (n = 312) and South Asians (n = 168) participating in STORK G who were genotyped and had fasting glucose data recorded. DNA samples were bisulfite converted using EZ DNA MethylationTM Kit (Zymo Research, Tustin, CA, USA) before added onto Infinium MethylationEPIC BeadChip (Illumina, San Diego, CA, USA) at the Department of Clinical Sciences, Clinical Research Centre, Lund University, Malmö, Sweden. Raw signal intensities of each probe were extracted using Illumina's GenomeStudio Software. The methylation level at each site was represented as a beta (β) value of the fluorescent intensity radio ranging from 0 (not methylated) to 1 (completely methylated). Meffil R package [51] (https://cran.r-project.org/) was used for quality control, normalization and reporting of beta values. During QC, we removed 8 samples: 1 due to sex mismatch (predicted sex outliers > 5SD), 1 outlier in control probes bisulfite 1 and bisulfite 2 (>5 SD), and 6 outliers from the methylated/unmethylated ratio comparison (>3 SD). Furthermore, 1299 probes with a detection p-value <0.01, and bead count <3 were removed. We used functional normalization, adjusting for effects of different batches, plates, columns and rows. A total of 307 European and 165 South Asian women and 864 560 probes of the array passed the QC.

**Pyrosequencing.** Random samples of 30 women were selected for technical validation of four CpGs sites by bisulfite pyrosequencing. The four CpG sites were chosen from preliminary top associations with fasting glucose (cg08098128, cg14120215), 2-hour glucose (cg19327414) and BMI (cg17148978). DNA samples were first bisulphite converted per QIAGEN Bisulfite conversion protocol [52] using 500 ng of DNA. Short DNA sequences that contained the CpG site of interest were amplified by PCR using PyroMark PCR kit from QIAGEN following the manufacturer instructions [53]. The PCR was performed on the 30 samples in duplicates with two positive controls, one was unmethylated converted DNA and the other was fully methylated converted DNA, both controls were commercially available from the EpiTect PCR Control DNA Set (100) by QIAGEN, and were used per the provider instructions. Also, a negative control only containing RNAse free water was used. Pyrosequecing was performed using the PyroMark Q48 Autoprep per the instructions provided in the user manual [54]. We used Bland-Altman plots to evaluate the agreement between pyrosequencing and the Infinium MethylationEPIC BeadChip, which have been used previously to assess agreement for this type of data [55,56]. To asses if there were proportional bias, we regressed the mean difference with the mean between methods. We also performed Pearson correlations between methods per CpG site and we followed a

previous published approach which consist in pooling all the CpG sites for the correlation analysis [57].

**Follow- up study of the women 10–12 years after delivery.** A 10-year follow-up of the women who attended STORK G is currently ongoing and expected to finish in 2021. The main aims are to assess the incidence of prediabetes and T2D and explore changes in risk factors for T2D (and CVD) over the last 10 years. We measure weight, height, physical inactivity, and blood pressure, and collect data on self-reported smoking, and chronic diseases/conditions. Currently, dried blood spots are biobanked and about 60% meet for fasting blood samples, and DNA from buccal swabs is being collected as well. We estimate to reach a sample of 350 women– 50% of those eligible.

## Results to date

In the EPIPREG sample, we excluded women without fasting plasma glucose available in week 28±2, and those without genotype data due to low DNA concentrations or problems with DNA extraction. For Europeans, we were able to quantify DNA methylation in 99% of the eligible samples (empty wells = 2 and full plates = 1), 87.2% of the total number of Europeans participating in STORK G at week 28±2. For South Asians, methylation status could be determined in 100% of the eligible subjects, representing 87.5% of total South Asians participating in STORK G at week 28±2 (Fig 1). Hence, EPIPREG resulted in DNA methylation data of 312 Europeans and 168 South Asians.

Some characteristics of the mothers and offspring of the EPIPREG sample are shown in Table 1. When comparing the clinical characteristics of the women and their offspring with and without DNA methylation data, the average of sedentary hours and percentage of truncal fat were significantly higher in the European subjects included in EPIPREG, whereas gestational week, hours of light and moderate-intense physical activity, and 25-hydroxyvitamin D levels were higher in the excluded samples (S1 Table). The differences were small and generally followed the same trend as the overall STORK G missing data analysis [34]. In South Asians we did not detect any significant differences between the women and their offspring included in EPIPREG vs the excluded individuals (S2 Table).

Genetic principal components distinctly separated Europeans and South Asian women (Fig 2), which fully corresponded with the self-reported ethnicity. South Asians were separated in two groups, the largest cluster mainly consisted of Pakistani women and the smaller group of Sri-Lankan women.

To assess the agreement between the Infinium MethylationEPIC BeadChip and the technical validation by pyrosequencing, we used Bland-Altman plots to illustrate agreement per CpG site (Fig 3). In Fig 3, we can see that most of the samples in cg17148978, cg14120215 and cg02098128 were within the limits of agreement (LAG). For cg17148978 the average mean difference was 0.74% (lower LAG:-0.56, upper LAG: 2.05%), for cg14120215 7.56% (lower LAG: -6.95, upper LAG: -22.09%) and for cg02098128–0.66% (lower LAG:-2.09, upper LAG 0.78%). We found no evidence of proportional bias in cg17148978, cg14120215 or cg02098128, meaning that the agreement in these CpG sites was good. In cg19327414, the mean difference was 7.79% (lower LAG: -12.32, upper LAG: 27.89%), and there was a significant proportional bias (Beta = -1.85, pval = <0.001). All the samples but one were within the lines of agreement. In the correlation analysis, the overall correlation when pooling all 4 CpG sites was high (r = 0.98, p<0.001) (Fig 4), while site specific correlations for cg17148978 (R = -0.23, p = 0.22), cg19327414 (R = 0.18, p = 0.40), cg14120215 (R = 0.18, p = 0.33) and cg02098128 (R = 0.30, p = 0.11) were weak.

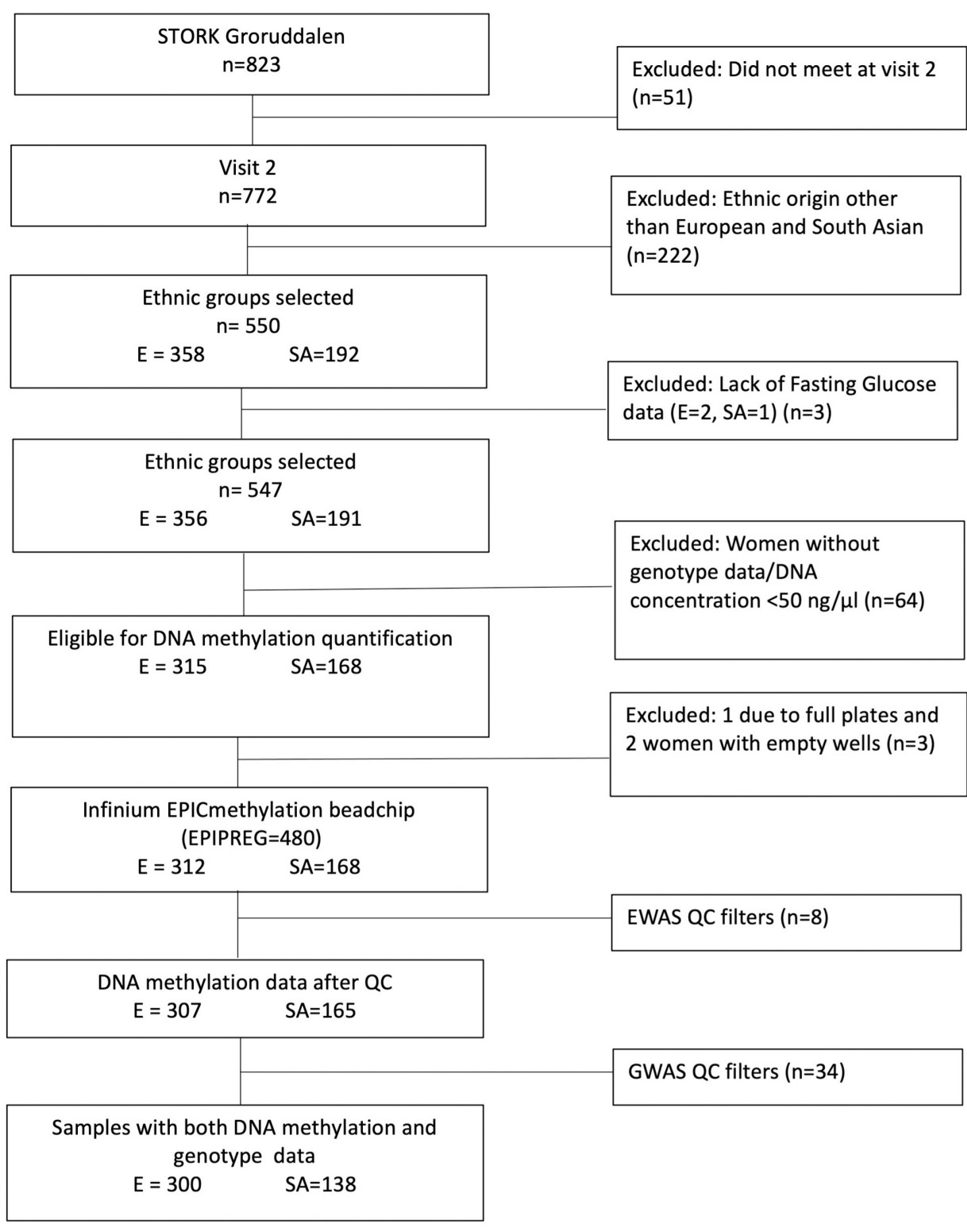

**Fig 1. Flow chart of the EPIPREG sample.** E = European, SA = South Asian.

**Table 1. Characteristics of the EPIPREG sample.**

| Variable | N | Europeans, n = 312 | South Asians, n = 168 |
|---|---|---|---|
| Age | 480 | 30.1 (4.6) | 28.2 (4.6) |
| Weeks' gestation | 480 | 28.1 (1.2) | 28.2 (1.2) |
| Height (cm) | 480 | 167.4 (5.8) | 159.9 (5.7) |
| Smoking status | 476 | | |
| Current | | 20 (6.5) | 1 (0.6) |
| 3 months pre-pregnancy | | 80 (25.9) | 2 (1.2) |
| Former* | | 87 (28.2) | 9 (5.4) |
| Never | | 122 (39.5) | 155 (92.8) |
| Pre-pregnancy alcohol intake, n (%) | 472 | 234 (76.2) | 5 (3.0) |
| Pre-pregnancy BMI (kg/m$^2$) | 474 | 24.6 (4.9) | 23.8 (4.1) |
| Actual BMI (kg/m$^2$) | 478 | 27.8 (4.7) | 26.8 (4.1) |
| Total fat (%) | 465 | 29.6 (9.7) | 26.2 (8.2) |
| Truncal fat (%) | 465 | 15.9 (5.5) | 14.0 (5.4) |
| Systolic blood pressure (mmHg) | 480 | 107.0 (9.6) | 101.1 (8.7) |
| Diastolic blood pressure (mmHg) | 480 | 68.4 (7.1) | 66.1 (6.9) |
| First degree relative with diabetes | 474 | 43 (14.0) | 79 (47.3) |
| GDM (WHO$_{2013}$), n (%) | 480 | 76 (24.4) | 70 (41.7) |
| GDM (WHO$_{1999}$), n (%) | 478 | 37 (11.9) | 25 (15.1) |
| Fasting glucose (mmol/L) | 480 | 4.7 (0.6) | 5.0 (0.6) |
| 2 hour glucose (mmol/L) | 477 | 6.0 (1.4) | 6.4 (1.5) |
| HbA1c (%) | 475 | 5.1 (0.3) | 5.3 (0.3) |
| Fasting Insulin (pmol/L) | 474 | 48.0 [33.0, 70.2] | 71.0 [57.0, 100.5] |
| Fasting C-peptide (pmol/L) | 474 | 712.0 [560.2, 901.8] | 855.5 [688.0, 1067.5] |
| HOMA-B | 474 | 173.2 [151.3, 199.7] | 179.1 [154.9, 207.9] |
| HOMA-IR | 474 | 1.5 [1.2, 1.9] | 1.8 [1.4, 2.3] |
| Total cholesterol (mmol/L) | 480 | 6.4 (1.1) | 6.0 (1.0) |
| HDL-cholesterol (mmol/L) | 480 | 1.9 (0.4) | 1.9 (0.4) |
| LDL-cholesterol (mmol/L) | 473 | 3.7 (1.0) | 3.3 (0.9) |
| Triglycerides (mmol/L) | 480 | 2.0 (0.7) | 2.0 (0. |
| Folate (nmol/l) | 471 | 13.0 [9.9, 18.0] | 12.0 [9.4, 16.0] |
| Vitamin D (μmol/L) | 472 | 69.9 (27.8) | 45.6 (22.2) |
| Leptin (μmol/L) | 476 | 1599.5 [966.9, 2497.7] | 2276.9 [1532.8, 3295.8] |
| Folate (nmol/l) | 471 | 13.0 [9.9, 18.0] | 12.0 [9.4, 16.0] |
| Vitamin B12 (μmol/L) | 472 | 211.0 [169.2, 253.0] | 186.0 [150.0, 232.2] |
| Sedentary time (hours/day)** | 382 | 18 (1.6) | 17.8 (1.7) |
| Light physical activity (hours/day)** | 382 | 4.3 (1.3) | 4.6 (1.3) |
| Moderate-intense physical activity (hours/day)** | 382 | 1.0 [0.7, 1.5] | 0.8 [0.5, 1.2] |
| Offspring data | | | |
| Gestational age (days) | 475 | 280.9 (11.7) | 277.1 (12.6) |
| Female sex, n (%) | 462 | 147 (49.0) | 79 (48.8) |
| Birth weight (g) | 472 | 3582.2 (529.4) | 3211.7 (512.2) |
| Birth length (cm) | 432 | 50.1 (2.3) | 49.3 (2.2) |
| Neonatal sum of skinfolds (mm) | 359 | 18.8 (4.0) | 17.0 (3.6) |

Data from gestational visit 2 for cross-sectional associations with DNA methylation data, otherwise specified. Data are presented in mean (SD) for normally distributed variables and median [IQR] for non-normal variables. Categorical variables are presented by frequency (%). Despite 8 individuals did not pass the QC procedure, the data of the 480 individuals are presented for informative purposes. WHO = World health organization.

GDM = gestational diabetes mellitus. BMI = body mass index. HDL = high-density lipoproteins. LDL = low-density lipoproteins.

* Ex-smokers and occasional smokers who did not smoke 3 months before pregnancy,

** Women with at least one valid day of registered physical activity (Armband).

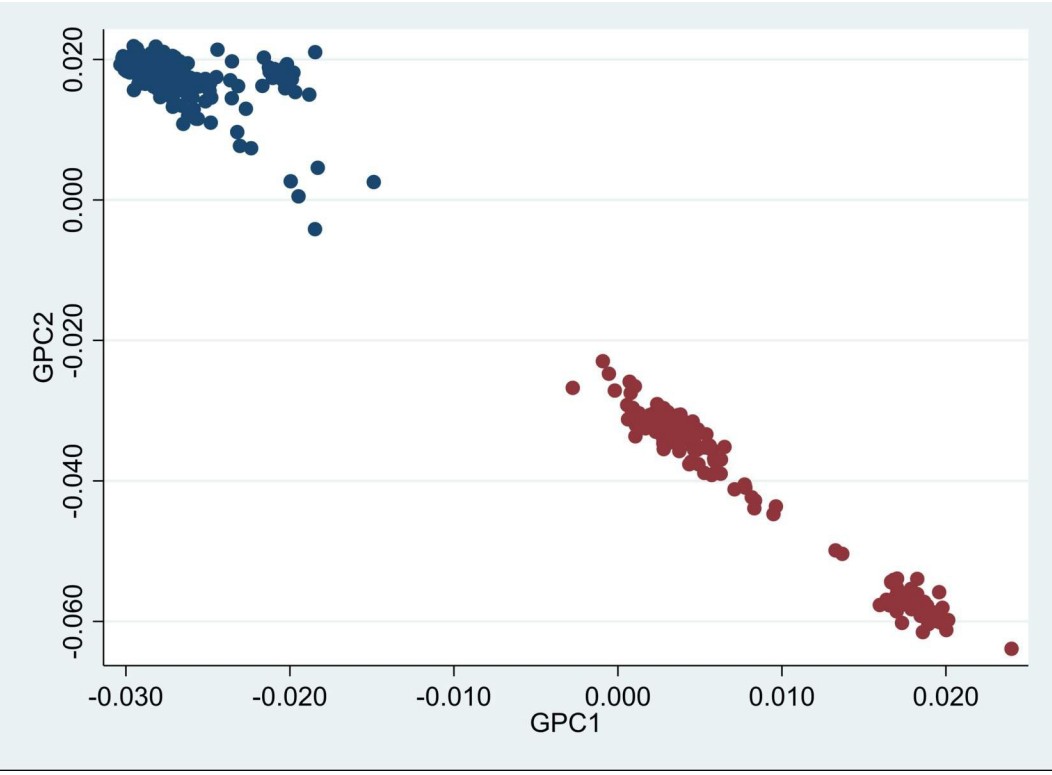

**Fig 2. Scatter dot plot of genetic PC1 (GPC1) and PC2 (GPC2) (n = 438).** Blue dots are Europeans, red dots are South Asians, based on self-reported ethnicity. It can be noticed that South Asians were separated in two groups, being the upper largest group mainly composed of Pakistanis and the smaller lower group of Sir Lankans.

## Strengths and limitations

EPIPREG is a population-based sample of European and South Asian pregnant women with epigenome-wide DNA methylation in maternal peripheral blood leukocytes. EPIPREG has detailed phenotype data from both the mother and the offspring, as well as genotype data. DNA methylation measured with the epigenome-wide chip showed high agreement with bisulphite pyrosequencing.

The inclusion of women with both European and South Asian ethnic background enables interesting studies into the role of DNA methylation in ethnic disparities in health. Furthermore, EPIPREG has both genome-wide genotype and DNA methylation data also allowing for methylation quantitative trait loci (mQTL) analysis.

Regarding the technical validation, correlations were low for each CpG site separately. However, correlations could be misleading for agreement analyses as they mainly measure the linearity of the variables irrespective of the data's shape [58], and are sensitive to the range of values—the broader the range, the higher the correlation coefficient [59]. Bland-Altman plots are considered a better test of agreement between methods [59], especially when the range of values is low as for three of our four CpG sites. Therefore although our individual sites showed low correlation between the two methods, they had high agreement.

EPIPREG's population-based design and comprehensive phenotyping allows for gaining representative data about the associations between DNA methylation and a wide range of phenotypic traits, exposures and outcomes. Hence, a major advantage of the cohort is the availability of several maternal phenotypes collected in gestational weeks 15 and 28, and 3 months

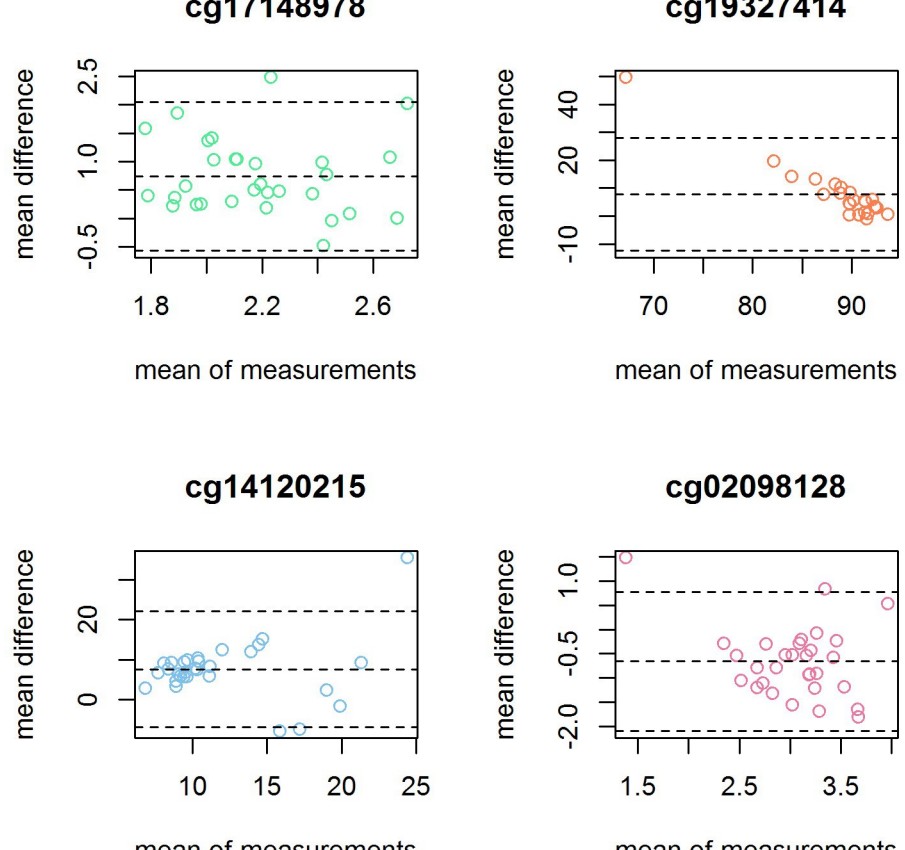

**Fig 3. Bland-Altman plots showing the mean difference between methods (y-axis) versus the mean between methods (x-axis) for each CpG site tested for technical validation.**

after delivery. Furthermore, there is an ongoing follow-up in some women 10 to 12 years after pregnancy. In the offspring we have anthropometric data recorded in utero, at birth and during the first four years of life, as well as serum cord blood and placental tissue biobanked. Lastly, we have permission for linkage with Norwegian national registries using the personal identification number.

Since EPIPREG has a moderate sample size, our study has limited statistical power for EWAS, nevertheless our broad availability of phenotypes will allow us to perform several DNA methylation-phenotype association analyses. Also, our sample is well suited for meta-analysis efforts, or to serve as a replication cohort. Another limitation is that DNA methylation is only measured in gestational week 28±2.

## Collaboration

EPIPREG may serve as a useful sample for generation of new hypotheses about associations between DNA-methylation and phenotypic traits relating to GDM, for replication of findings from other studies, and for meta-analysis efforts. We are currently welcoming collaborations with cohorts with similar data and researchers interested in collaboration are welcome to contact Christine Sommer, or visit our website: www.epipreg.no.

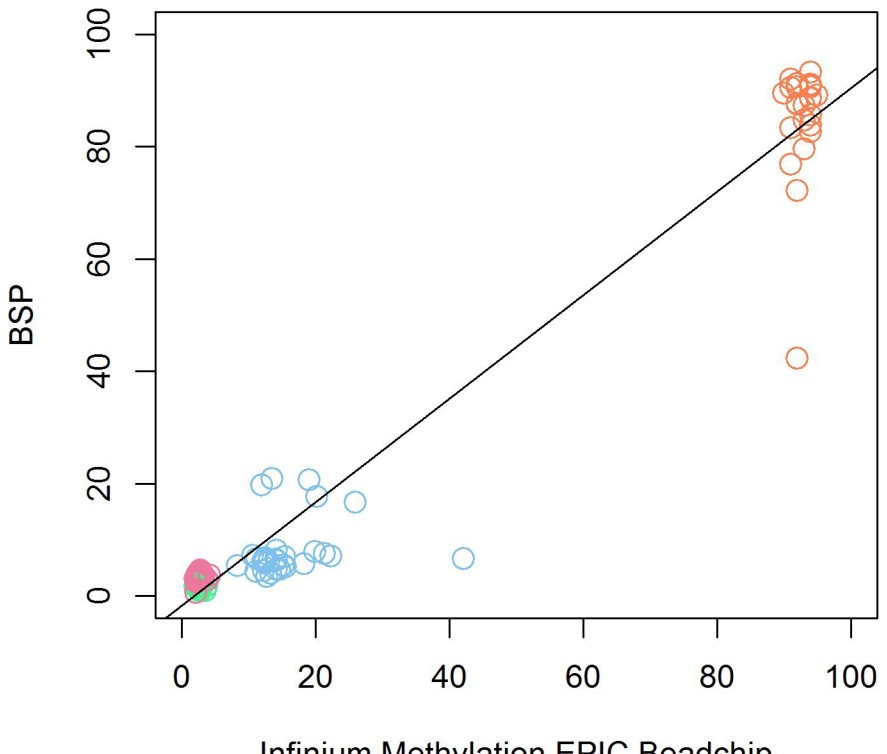

**Fig 4. Scatter plot showing the relationship between the DNAm values of the four selected CpG sites quantified with the Infinium MethylationEPIC BeadChip (x-axis) and with bisulphite pyrosequencing (BSP) (y-axis).** Dots colour code: Pink: cg02098128, green: cg17148978, blue: cg14120215, orange: cg19327414.

## Supporting information

**S1 Table. Mean comparison between the European samples included in the Infinium MethylationEPIC BeadChip versus their respective excluded samples.**
(XLSX)

**S2 Table. Mean comparison between the South Asian samples included in the Infinium MethylationEPIC BeadChip versus their respective excluded samples.**
(XLSX)

## Acknowledgments

We would like to thank the women who participated in the STORK Groruddalen study, Maria Sterner, Malin Neptin, and Gabriella Gremsperger at the Genomics Diabetes and Endocrinology CRC, Malmö, for experiments.

## Author Contributions

**Conceptualization:** Anne Karen Jenum, Kåre I. Birkeland, Christine Sommer.

**Data curation:** Sindre Lee-Ødegård, Kåre Rønn Richardsen, Line Sletner.

**Formal analysis:** Nicolas Fragoso-Bargas, Julia O. Opsahl, Nadezhda Kiryushchenko, Gunn-Helen Moen, Christine Sommer.

**Funding acquisition:** Elisabeth Qvigstad, Christine Sommer.

**Investigation:** Nicolas Fragoso-Bargas, Christine Sommer.

**Methodology:** Nicolas Fragoso-Bargas, Julia O. Opsahl, Nadezhda Kiryushchenko, Yvonne Böttcher, Sindre Lee-Ødegård, Kåre Rønn Richardsen, Christin W. Waage, Line Sletner, Anne Karen Jenum, Rashmi B. Prasad, Leif C. Groop, Gunn-Helen Moen, Kåre I. Birkeland, Christine Sommer.

**Project administration:** Christine Sommer.

**Supervision:** Christine Sommer.

**Visualization:** Nicolas Fragoso-Bargas, Nadezhda Kiryushchenko, Christine Sommer.

**Writing – original draft:** Nicolas Fragoso-Bargas.

**Writing – review & editing:** Julia O. Opsahl, Nadezhda Kiryushchenko, Yvonne Böttcher, Sindre Lee-Ødegård, Elisabeth Qvigstad, Kåre Rønn Richardsen, Christin W. Waage, Line Sletner, Anne Karen Jenum, Rashmi B. Prasad, Leif C. Groop, Gunn-Helen Moen, Kåre I. Birkeland, Christine Sommer.

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
