## [Decision Letter · Decision Letter 0]

24 May 2021

PONE-D-21-08932

Cohort Profile: Epigenetics in Pregnancy (EPIPREG) – population-based sample of European and South Asian pregnant women with epigenome-wide DNA methylation (850k) in peripheral blood leukocytes.

PLOS ONE

Dear Dr. Sommer,

Thank you for submitting your manuscript to PLOS ONE. After careful consideration, we feel that it has merit but does not fully meet PLOS ONE’s publication criteria as it currently stands. Therefore, we invite you to submit a revised version of the manuscript that addresses the points raised during the review process.

We look forward to receiving your revised manuscript.

Kind regards,

Lee-Ling Lim

Academic Editor

PLOS ONE

Additional Editor Comments:

Please address all reviewer comments.

Journal Requirements:

3. We note that you are reporting an analysis of a microarray, next-generation sequencing, or deep sequencing data set. PLOS requires that authors comply with field-specific standards for preparation, recording, and deposition of data in repositories appropriate to their field. Please upload these data to a stable, public repository (such as ArrayExpress, Gene Expression Omnibus (GEO), DNA Data Bank of Japan (DDBJ), NCBI GenBank, NCBI Sequence Read Archive, or EMBL Nucleotide Sequence Database (ENA)). In your revised cover letter, please provide the relevant accession numbers that may be used to access these data. For a full list of recommended repositories, see http://journals.plos.org/plosone/s/data-availability#loc-omics or http://journals.plos.org/plosone/s/data-availability#loc-sequencing.

4. Thank you for stating the following in the Financial Disclosure section:

[EPIPREG is supported by the South Eastern Norway Regional Health Authority (grant number: 2019092), and the Norwegian Diabetes Association (grant number: N/A). G.H.M. is supported by the Norwegian Research Council (Post doctoral mobility research grant 287198), and have received funding support by Nils Normans minnegave (grant number: N/A)]. 

We note that you received funding from a commercial source: Nils Normans minnegave

Reviewers' comments:

Reviewer's Responses to Questions

**Comments to the Author**

1. Is the manuscript technically sound, and do the data support the conclusions?

Reviewer #1: Partly

Reviewer #2: Yes

3. Have the authors made all data underlying the findings in their manuscript fully available?

Reviewer #1: No

Reviewer #2: No

4. Is the manuscript presented in an intelligible fashion and written in standard English?

Reviewer #1: Yes

Reviewer #2: Yes

5. Review Comments to the Author

Reviewer #1: This paper describes a new cohort with pregnancy data on a set of women from two different ethnic groups.

The cohort is of interest however I am disappointed that the authors have not performed any basic analysis with any of the cardio-metabolic traits or other pregnancy related traits. Moreover, none of the methods (experimental nor statistical analysis) are novel therefore this is merely a report of a new collected cohort with some description of demographics.

Reviewer #2: This is an excellent presentation of a very interesting cohort. I have some minor comments which may help to improve the manuscript content.

The authors state several times that the cohort is of a large size although they do also say they have limited statistical power in their limitations section. I would argue the cohort is of modest size, however the quantity and range of data collected on each mother-child pair is sizable and the repeated measures are a major advantage to the cohort design.

Within the introduction the paper would benefit from a justification of why there is value in a cohort collection with 2 ethnic groups.

Additional minor comments:

Suggest referring to the EPIC array as Infinium MethylationEPIC BeadChip kit rather than just MethylationEPIC kit to make the resource more discoverable.

In the introduction there are other citations that the authors could consider including and which support the content presented. For example, there are other large studies showing the association between T2D and methylation eg Juvinao-Quintero DL et al (DOI: 10.1186/s13148-021-01027-3) and some studies which have a multi-ethnic study design relevant to EPIPREG eg Chambers JC et al (doi: 10.1016/S2213-8587(15)00127-8). For smoking citations (#16 or #17) it might be also useful to include eg Joehanes R et al (10.1161/CIRCGENETICS.116.001506), Wiklund P et al (DOI: 10.1186/s13148-019-0683-4) and/or Joubert et al (DOI: 10.1289/ehp.1205412). For alcohol intake there are other published papers that are relevant eg Dugue AP (blood)(DOI: 10.1111/adb.12855) and Xu K (although this is a study in saliva not blood)(DOI: 10.1111/acer.14168). Citation #8 is a review so should be acknowledged as such.

“The population based design and inclusion of a significant number of women with European and South Asian ethnicity allows us to study a wide range of phenotypes.” I would suggest removing the words “significant number” from this sentence as it is subjective.

The introduction is clearly written. It could however be improved by discussing the rationale for inclusion of women of European and South Asian ancestries.

In the “Study Population” section it would be more informative to report the specific participation rates in the two ethnic groups in EPIPREG rather than the range across all groups in STORK G.

Line 134: “…if the last was born outside Europe” suggest changing to “if the latter was born outside Europe”

Line 164: A citation to the Oxford University HOMA Calculator should be included.

Line 173: “intense?” typo ?

Line 186 (and 189): suggest amending “Formalin-fixated- paraffin embedded blocks” to “Formalin-Fixed Paraffin-Embedded (FFPE) blocks”

Line 215: for imputation of genetic data it isn’t clear if imputation was conducted in each ethnic group separately or not (or what reference panel was used).

Line 246: typo “QUIAGEN” -> “QIAGEN”

Line 247: suggest replacing the work “doublet” with “duplicate” since this is a more common way of describing replicates.

Line 248: It is unclear what the unmethylated and methylated controls are. Presumably these are commercially available samples(?) It needs to be made clear the negative control is a sample containing no DNA template (if this is the case).

Line 298: The R2 of 0.98 is misleading as it doesn’t actually tell us anything about correlation between methods on a per CpG basis. There should be 4x R2 measures reported, one for each CpG tested which would give an estimate of agreement between methods for each CpG.

Line 309-310: There are a number of other studies studying epigenetics perinatally (eg members of the PACE consortium https://www.niehs.nih.gov/research/atniehs/labs/epi/pi/genetics/pace/index.cfm) and cohorts who also have South Asian and European ancestry maternal and offspring samples eg Born in Bradford: https://borninbradford.nhs.uk/). I would argue that in terms of sample size EPIPREG is relatively small compared to some of these other cohorts.

Figure 2: There appears to be some population stratification which is particularly noticeable in the South Asian group (two groups are evident on both the 1st and 2nd PCs). It would be useful to comment if this can be explored further.

Figure 3: This figure isn’t very informative given that the CpG sites tested have very different distributions. A 4 panel figure showing the R2 for each of the 4 CpG sites would show the relationship between the two methods tested much more clearly. This plot could also be improved if the scales were labelled the same (ie both 0-1 or 0-100) and the labels were descriptive (eg “% methylation measured by bisulphite pyrosequencing” instead of “BSP”).

6. PLOS authors have the option to publish the peer review history of their article (what does this mean?). If published, this will include your full peer review and any attached files.

Reviewer #1: No

Reviewer #2: No

2. Has the statistical analysis been performed appropriately and rigorously? 

Reviewer #2: Yes

---

## [Author Response · Author response to Decision Letter 0]

3 Jul 2021

Response to reviewers

Reviewer #1: 

This paper describes a new cohort with pregnancy data on a set of women from two different ethnic groups.

The cohort is of interest however I am disappointed that the authors have not performed any basic analysis with any of the cardio-metabolic traits or other pregnancy related traits. Moreover, none of the methods (experimental nor statistical analysis) are novel therefore this is merely a report of a new collected cohort with some description of demographics.

Reply = Thank you for your comment! We did not include any association analyses of cardio-metabolic related traits in this manuscript, since the aim of this paper is to present the EPIPREG sample with epigenetics, genetics and a variety of interesting phenotypes. We noticed that this aim was only mentioned in the abstract, but have now clarified it and added the aim also to the introduction (lines 111-113 in Manuscript with track changes). 

Reviewer #2: 

This is an excellent presentation of a very interesting cohort. I have some minor comments which may help to improve the manuscript content.

The authors state several times that the cohort is of a large size although they do also say they have limited statistical power in their limitations section. I would argue the cohort is of modest size, however the quantity and range of data collected on each mother-child pair is sizable and the repeated measures are a major advantage to the cohort design.

Reply = Thanks for pointing this out, we have corrected the terminology and stated that the cohort has a moderate sample size throughout the manuscript.

Within the introduction the paper would benefit from a justification of why there is value in a cohort collection with 2 ethnic groups.

Reply = We have added a few lines to the introduction highlighting the importance of having two ethnic groups (lines 103-108).

Additional minor comments:

Suggest referring to the EPIC array as Infinium MethylationEPIC BeadChip kit rather than just MethylationEPIC kit to make the resource more discoverable.

Reply= The modification was done

In the introduction there are other citations that the authors could consider including and which support the content presented. For example, there are other large studies showing the association between T2D and methylation eg Juvinao-Quintero DL et al (DOI: 10.1186/s13148-021-01027-3) and some studies which have a multi-ethnic study design relevant to EPIPREG eg Chambers JC et al (doi: 10.1016/S2213-8587(15)00127-8). For smoking citations (#16 or #17) it might be also useful to include eg Joehanes R et al (10.1161/CIRCGENETICS.116.001506), Wiklund P et al (DOI: 10.1186/s13148-019-0683-4) and/or Joubert et al (DOI: 10.1289/ehp.1205412). For alcohol intake there are other published papers that are relevant eg Dugue AP (blood)(DOI: 10.1111/adb.12855) and Xu K (although this is a study in saliva not blood)(DOI: 10.1111/acer.14168). Citation #8 is a review so should be acknowledged as such.

Reply= Thanks for recommending these studies; they were added in the introduction, but Xu study was not included since we limited our scope to blood. Since we mainly used research papers for this part in the introduction, in citation 8 we replaced the review with a recently published EWAS of GDM (https://doi.org/10.2337/dc20-2960). 

“The population based design and inclusion of a significant number of women with European and South Asian ethnicity allows us to study a wide range of phenotypes.” I would suggest removing the words “significant number” from this sentence as it is subjective.

Reply= As suggested, we removed “of a significant number” in all the manuscript

The introduction is clearly written. It could however be improved by discussing the rationale for inclusion of women of European and South Asian ancestries.

Reply= Thank you! We have added it to the introduction (lines 103-108).

In the “Study Population” section it would be more informative to report the specific participation rates in the two ethnic groups in EPIPREG rather than the range across all groups in STORK G.

Reply= The participation rates of Europeans and South Asians that participated in STORK G has been added (line 126-127).

Line 134: “…if the last was born outside Europe” suggest changing to “if the latter was born outside Europe”

Reply= Corrected

Line 164: A citation to the Oxford University HOMA Calculator should be included.

Reply= The citation was added

Line 173: “intense?” typo ?

Reply= Typo erased

Line 186 (and 189): suggest amending “Formalin-fixated- paraffin embedded blocks” to “Formalin-Fixed Paraffin-Embedded (FFPE) blocks”

Reply= Corrected

Line 215: for imputation of genetic data it isn’t clear if imputation was conducted in each ethnic group separately or not (or what reference panel was used).

Reply= We have now added that the imputation was done separately in EUR and SA, and that we used the 1000 genome panels specific for Europeans and South Asians populations (Lines 230-237).

Line 246: typo “QUIAGEN” -> “QIAGEN”

Reply= Typo corrected

Line 247: suggest replacing the work “doublet” with “duplicate” since this is a more common way of describing replicates.

Reply= Doublet was replaced with duplicate

Line 248: It is unclear what the unmethylated and methylated controls are. Presumably these are commercially available samples(?) It needs to be made clear the negative control is a sample containing no DNA template (if this is the case).

Reply= Specifications of the control used were added (Lines 271-274). 

Line 298: The R2 of 0.98 is misleading as it doesn’t actually tell us anything about correlation between methods on a per CpG basis. There should be 4x R2 measures reported, one for each CpG tested which would give an estimate of agreement between methods for each CpG.

Reply= Thank you for your comment! We originally followed the approach from Ronn et al 2013 (https://doi.org/10.1371/journal.pgen.1003572), where they pooled their sites together for the correlation analysis. As suggested we have added correlation per CpG site. We have also added Bland-Altman plots as presented by Mamtani and colleagues (https://doi.org/10.1186/s13148-016-0173-x), and regressed the mean difference and the average of EPIC and pyrosequencing values to asses if there were proportional bias. We have now added some lines to the methods and results (lines 276-281 and 334-347).

Correlations indeed have some limitations in statistical testing of agreement, especially when there is a small range of values/small variation as we have for 3 of 4 CpG sites, https://doi.org/10.1159/000337798), while Bland-Altman plots illustrate agreement better (https://doi.org/10.1038/ki.2008.306). We have added a few lines to the Strengths and limitations section discussing the results (lines 361-367).

Line 309-310: There are a number of other studies studying epigenetics perinatally (eg members of the PACE consortium https://www.niehs.nih.gov/research/atniehs/labs/epi/pi/genetics/pace/index.cfm) and cohorts who also have South Asian and European ancestry maternal and offspring samples eg Born in Bradford: https://borninbradford.nhs.uk/). I would argue that in terms of sample size EPIPREG is relatively small compared to some of these other cohorts.

Reply= We have changed the wording to moderate throughout the paper. 

Figure 2: There appears to be some population stratification which is particularly noticeable in the South Asian group (two groups are evident on both the 1st and 2nd PCs). It would be useful to comment if this can be explored further.

Reply=Thanks, for pointing this out! We are aware and have added a few lines mentioning the reason for the stratification (lines 329-331).

Figure 3: This figure isn’t very informative given that the CpG sites tested have very different distributions. A 4 panel figure showing the R2 for each of the 4 CpG sites would show the relationship between the two methods tested much more clearly. This plot could also be improved if the scales were labelled the same (ie both 0-1 or 0-100) and the labels were descriptive (eg “% methylation measured by bisulphite pyrosequencing” instead of “BSP”).

Reply=We have added a four-panel figure showing the agreement between the chip and pyrosequencing using Bland-Altman plots.

---

## [Decision Letter · Decision Letter 1]

2 Aug 2021

Cohort Profile: Epigenetics in Pregnancy (EPIPREG) – population-based sample of European and South Asian pregnant women with epigenome-wide DNA methylation (850k) in peripheral blood leukocytes.

PONE-D-21-08932R1

Dear Dr. Sommer,

We’re pleased to inform you that your manuscript has been judged scientifically suitable for publication and will be formally accepted for publication once it meets all outstanding technical requirements.

Kind regards,

Lee-Ling Lim

Academic Editor

PLOS ONE

Additional Editor Comments (optional):

Reviewers' comments:

Reviewer's Responses to Questions

**Comments to the Author**

1. If the authors have adequately addressed your comments raised in a previous round of review and you feel that this manuscript is now acceptable for publication, you may indicate that here to bypass the “Comments to the Author” section, enter your conflict of interest statement in the “Confidential to Editor” section, and submit your "Accept" recommendation.

Reviewer #2: All comments have been addressed

2. Is the manuscript technically sound, and do the data support the conclusions?

Reviewer #2: Yes

3. Has the statistical analysis been performed appropriately and rigorously? 

Reviewer #2: Yes

4. Have the authors made all data underlying the findings in their manuscript fully available?

Reviewer #2: Yes

5. Is the manuscript presented in an intelligible fashion and written in standard English?

Reviewer #2: Yes

6. Review Comments to the Author

Reviewer #2: The authors have made revisions which satisfy the comments/questions raised in my original review. I have no further comments and recommend this paper for publication.

7. PLOS authors have the option to publish the peer review history of their article (what does this mean?). If published, this will include your full peer review and any attached files.

Reviewer #2: No

---

## [Editor Report · Acceptance letter]

5 Aug 2021

PONE-D-21-08932R1 

Cohort Profile: Epigenetics in Pregnancy (EPIPREG) – population-based sample of European and South Asian pregnant women with epigenome-wide DNA methylation (850k) in peripheral blood leukocytes 

Dear Dr. Sommer:

I'm pleased to inform you that your manuscript has been deemed suitable for publication in PLOS ONE. Congratulations! Your manuscript is now with our production department. 

Kind regards, 

on behalf of

Dr. Lee-Ling Lim 

Academic Editor

PLOS ONE